# SARS-CoV-2 protein structure and sequence mutations: Evolutionary analysis and effects on virus variants

Ugo Lomoio[1]☯, Barbara Puccio[1]☯, Giuseppe Tradigo[2], Pietro Hiram Guzzi[1]*, Pierangelo Veltri[3]

**1** Department of Surgical and Medical Sciences, University of Catanzaro, Catanzaro, Italy, **2** e-Campus University, Novedrate, Italy, **3** DIMES, University of Calabria, Rende, Italy

☯ These authors contributed equally to this work.
* hguzzi@unicz.it

**Data Availability Statement:** The repository \url {https://github.com/UgoLomoio/SARSCoV2_ variants_PCN} contains data, code, and additional figures used in this work.

## Abstract

The structure and sequence of proteins strongly influence their biological functions. New models and algorithms can help researchers in understanding how the evolution of sequences and structures is related to changes in functions. Recently, studies of SARS-CoV-2 Spike (S) protein structures have been performed to predict binding receptors and infection activity in COVID-19, hence the scientific interest in the effects of virus mutations due to sequence, structure and vaccination arises. However, there is the need for models and tools to study the links between the evolution of S protein sequence, structure and functions, and virus transmissibility and the effects of vaccination. As studies on S protein have been generated a large amount of relevant information, we propose in this work to use Protein Contact Networks (PCNs) to relate protein structures with biological properties by means of network topology properties. Topological properties are used to compare the structural changes with sequence changes. We find that both node centrality and community extraction analysis can be used to relate protein stability and functionality with sequence mutations. Starting from this we compare structural evolution to sequence changes and study mutations from a temporal perspective focusing on virus variants. Finally by applying our model to the Omicron variant we report a timeline correlation between Omicron and the vaccination campaign.

## Introduction

Comprehension of cellular processes requires studying relations between the sequence of genes and the structure of encoded proteins [1, 2] through genomic and proteomic studies. From an evolutionary point of view, changes in gene sequences (e.g. single nucleotide mutations) may imply a modification of protein structure and, thus, phenotype changes. The evolutionary process limits some phenotype modifications due to environmental constraints [1, 3].

The pandemic of the SARS-CoV-2 virus has led to the storage of a large amount of genomic and proteomic datasets enabling the molecular evolutionary analysis of proteins thereby

**Funding:** The authors received no specific funding for this work.

**Competing interests:** The authors have declared that no competing interests exist.

boosting studies of the relations between the protein sequence structure and functions of the virus [4, 5]. The genome of SARS-CoV-2 contains 29.9 kilobase [6], and has 14 functional open reading frames (ORFs) and multiple encoding regions: (i) four structural proteins (i.e., Spike, S; envelope, E; membrane, M; and nucleocapsid, N); (ii) 16 nonstructural proteins (nsp1-nsp16) and (iii) accessory proteins [5, 7, 8].

Viruses undergo many mutations during the process of replication [9, 10]. Mutations can occur randomly due to errors in replication steps and to changes in the structure of the viral proteins [11, 12]. These mutations may acquire an evolutionary advantage when they improve their ability to escape from the host's immune system [13]. Many studies have identified mutations in SARS-CoV-2 [5, 14–17], and in the relation between virus mutations and the ability to override immune systems (e.g., relating the evolutionary process with the spread of the virus).

Sequence of mutations may cause the insurgence of variants. A variant is a viral genome with one or more mutations causing changes in protein structure and virus characteristics. Numerous studies have focused on the evolution of SARS-CoV-2 main variants, highlighting the increase of transmissibility vis-a-vis a reduction in severity, (e.g. Omicron variant [17]). The World Health Organisation (WHO) and the European Center for Disease Control (ECDC) are tasked with studying new evidence on variants. Sequenced samples of SARS-CoV-2 gathered from around the world are analysed by the WHO and ECDC for publishing information about variants and health protocols. Variants and available information are published [18–20] and in the ECDC/WHO register available on line. Variants are organised into lineages, groups of closely related viruses with a common ancestor. Although mutations occur very frequently during virus replication, only few modifications change the virus functionalities. A group of variants with similar genetic changes has been designed by the WHO as a Variant Being Monitored (VBM), Variant of Concern (VOC), or as a Variant of Interest (VOI) due to shared attributes and characteristics that may require government action to safeguard public health [21, 22].

We focus on a subset of variants, responsible for S protein structure modifications, by analysing structural models gathered from the *Exascalate4Cov* consortium database [23]. We study the impact of sequence changes on Spike protein structure by means of the Protein Contact Network (PCN) formalism [24–28]. PCNs are graphs whose nodes represent the $C - \alpha$ atoms of the backbone of proteins, while edges represent a relative spatial distance of 4—8 Ångstroms among residues. Topological descriptors of PCNs, such as node centrality measures, are used to discover protein properties, even at the sub-molecular level [26, 27]. We focus on the correlation between sequence and structure evolution and figure out relations between sequence updates, structure and phenotype variations. A similar model has been used to study Spike proteins of variants of SARS-CoV-2. We investigate how sequence changes generate relevant changes in the structure. We calculate centrality measures for each PCN node and their changes due to mutations. We also measure the structural differences among variants using the template modelling score (TM-SCORE) [29], a metric for assessing the topological similarity of protein structures. Moreover, using the Louvain algorithm we extract communities for each PCN considered [30] and we then relate them to mutations. Finally we build two trees, one considering the difference in network descriptors and the second regarding the distance in terms of TM-SCORE to clarify possible divergences between the timeline evolution of sequences and structure. We analysed each variant from three perspectives: the sequence, the structure, and the PCN network parameters. The obtained results allowed us to put forward the following claims:

- The temporal evolution of the variants shows that both sequence and structure of Spike proteins changed significantly after the beginning of the large-scale vaccination campaign.

- The PCN analysis shows local changes in the protein structure of the studied variants, which can be related to protein folding and stability.

- PCN nodes centrality measures highlight the differences between Spike proteins in: (i) Omicron$_1$ variant versus Wild Type, and (ii) Delta variant versus Wild Type.

- Communities extracted with the Louvain algorithm represent correlations among amino acids which correspond to different functional domains of spike proteins, for example, with the Louvain algorithm communities can be mapped into the SARS-CoV2 Omicron$_1$ Spike variant structure.

- **For S proteins, the net charge of RBD and NTD domain of the S protein, predicted from $pK_a$ values, increases with time**.

In this paper we focus on: (i) the correlation between sequence changes and structural changes in the evolving Spike proteins of the SARS-CoV-2 variants; (ii) correlation between vaccination campaigns and Omicron variants mutations. Similar works [17, 31–33] covered the same problem: for example [17] focused on the impact of mutations on the virus pathogenicity, while [33] focused on the impact on binding affinity. In this paper, we also analyse parameters extracted from both sequence and structure, integrating different viewpoints. Unlike the works which consider different scales [32], this paper also analyses the impact of mutations at the intermediate level by using the PCN model [31] and considers a larger set of structures and parameters.

## Materials and methods

We here propose the analysis of sequence mutations and structural changes using the pipeline described in Fig 1. The structures of the Spike protein of some selected virus variants are used as input to the pipeline. PDB files of the three-dimensional Spike structures are gathered from *Exascalate4Cov* consortium database [23]. Table 1 reports the variants used as input in the pipeline and the related mutations on Spike protein.

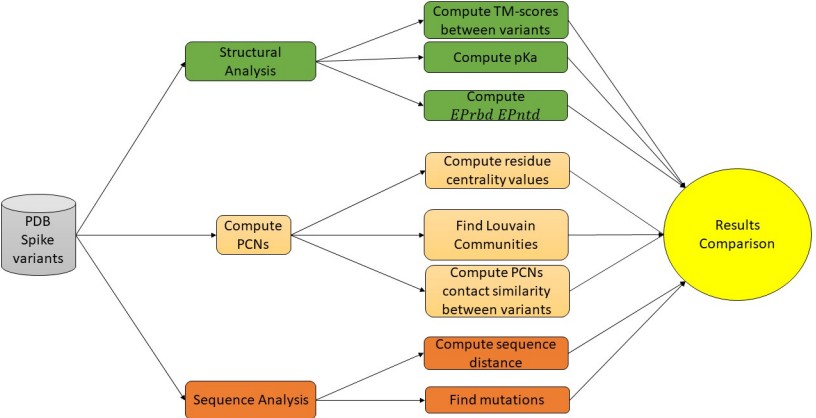

**Fig 1. Analysis pipeline.** Sequence and structure information related to Spike variants are extracted from the *Exascalate4Cov* Consortium Database. Structural analysis is performed by calculating differences using the TM-Score and Net charge of the Spike Protein. Protein Contact Networks search communities to find similarities among variants and to evaluate residue centrality values. Sequence analysis is studied to compute distance among sequences and, thus, between variants.

**Table 1. SARS-CoV-2 variants, lineage classification and mutations on the Spike protein sequence.**

| Variant | Lineage | Mutations on Spike Protein |
|---|---|---|
| Alpha ($\alpha$) | B.1.1.7 | HV69–70Δ, Y144Δ, N501Y, A570D, D614G, P681H, T716I, S982, D1118H |
| Beta ($\beta$) | B.1.351 | D80A, D215G, 241–243Δ, R246I, K417N, E484K, N501Y, D614G, A701V |
| Omicron$_1$ ($o_1$) | BA.1 | A67V, HV69–70Δ, T95I, G142D, V143Δ, YY144–145Δ, N211Δ, L212I, ins214EPE, G339D, S371L, S373P, S375F, K417N, N440K, G446S, S477N, T478K, E484A, Q493R, G496S, Q498R, N501Y, Y505H, T547K, D614G, H655Y, N679K, P681H, N764K, D796Y, N856K, Q954H, N969K, L981F |
| Omicron$_5$ ($o_5$) | BA.5 | Δ25–27, HV69–70Δ, G142D, V213G, G339D, S371F, S373P, S375F, T376A, D405N, R408S, K417N, N440K, L452R, S477N,T 478K, E484A, F486V, Q498R, N501Y, Y505H, D614G, H655Y, N679K, P681H, N764K, D796Y, Q954H, N969K |
| Gamma ($\gamma$) | P.1 | D138Y, R190S, K417T, E484K,N501Y, D614G, H655Y, T1027I |
| Zeta ($\zeta$) | P.2 | E484K, D614G, V1176F |
| Delta ($\delta$) | B.1.617.2 | EE156–157Δ, R158G, L452R, T478K, D614G, P681R, D950N |
| Kappa ($\kappa$) | B.1.617.1 | E154K, E484Q, D614G, P681R, Q1071H |
| Epsilon ($\epsilon$) | B.1.427 | S13I, W152C, L452R, D614G |
| Eta ($\eta$) | B.1.525 | Q52R, A67V, HV69–70Δ, Y144Δ, E484K, D614G, Q677H, F888L |
| Iota1$_1$ ($\iota_1$) | B.1.526 | L5F, T95I, D253G, S477N, D614G |
| Iota$_2$ ($\iota_2$) | B.1.526 | L5F, T95I, D253G, E484K, D614G |
| Ihu | B.1.640.1 | E96Q, CNDPFLGVY136–144Δ, R190S, I210T, R346S, N394S, Y449N, F490R, N501Y, D614G, P681H, T859N, D936H, K1191N |

For each Spike protein we compute a corresponding PCN using the PCN-Miner tool [31]. Each node of the PCN corresponds to a single amino acid of the corresponding protein, while an edge connects two nodes whose spatial distance is comprised of between 4 and 8 Angstroms [26]. For each node of the PCN a set of centrality measures is evaluated. We used the following: *Betweenness*, *Degree*, Eigenvector, *Closeness*, and *Katz*, centrality measures. A description and definition are reported in Table 2.

**Table 2. Centrality measures definition.**

| Measure Description | Measure Definition |
|---|---|
| *Betweenness Centrality measure*: given a node *i*, it measures how much the node (i.e., amino acid) influences communication and serves as a bridge from one part of the graph (i.e., part of the represented protein) to another. $\sigma_{j,k}$ indicates the number of shortest paths from node *j* to node *k* and $\sigma_{j,k}(i)$ indicates the shortest path which includes *i*. | $C_{betweennes}(i) = \sum_{i \neq j \neq k} \frac{\sigma_{j,k}(i)}{\sigma_{j,k}}.$ |
| *Degree Centrality measure*: given a node *i*, it measures the normalised degree of *i*, i.e. the number of connections of the node. Nodes (amino acids) with a high centrality, considered hubs in the network (i.e., the protein), have a crucial role in the network communication. The degree centrality of a node *i* is computed as reported on the right. | $C_{deg}(i) = \frac{deg(i)}{max(degrees)}$ |
| *Eigenvector Centrality measure*: given a node *i*, the Eigenvector centrality measure how a node is connected to other central nodes. A high value means that a node *i* is connected to many high-score nodes. Let A be the adjacency matrix of a graph G. The eigenvector centrality $C_{eig}$ of a node *i* is given by the formula $C_{eig}(i) = \frac{1}{\lambda} \sum_{t \in M_v} x_t = \frac{1}{\lambda} \sum_{t \in G} a_{v,t} x_t$, where $M_{(v)}$ is the set of neighbours and $\lambda$ is a constant. | $C_{eig}(i) = \frac{1}{\lambda} \sum_{j \in G} A(i,j) C_{eig}(j)$ |
| *Closeness Centrality measure*: given a node *i*, it measures the distance (closeness) among *i* (amino acid) to all graph nodes and it is evaluated as reported on the right, where $d(i, j)$ is the shortest distance between *i* and *j*. | $C_{closeness}(i) = \frac{n-1}{\sum_{j=1}^{j=n-1, j \neq i} d(i,j)}.$ |
| *Katz Centrality measure*: given a node *i*, it measures the influence degree of *i* in the graph (protein). $\alpha$ and $\beta$ (see right part) are parameters indicating (i) the attenuation factor and (ii) the weight attributed to the neighborhoods of each node. | $C_{katz}(i) = \sum_j \alpha A_{i,j} x_j + \beta$ |

## PCN analysis

We measure the centrality values of all the PCN nodes and compare both the overall changes (i.e., averaging the centrality values) and local changes (i.e., changes in the centrality values of mutated residues). We depict these values by using boxplots and radar plots. Boxplots are associated with all variants. Radar plots are used to represent centrality values of mutated nodes in the Spike variants. Finally, the obtained centrality measures are mapped onto the real protein structure using PCN-Miner. A t-test [34] on the variants centrality distribution is used to evaluate the significance of any changes. Community detection analysis has been performed with the Louvain algorithm with default parameters [30] to study the relation between virus mutations and communities in PCNs. The Louvain method allows us to find communities in graphs. Based on the greedy paradigm, which uses modularity graph information to optimise performance on large graphs, it works by finding small communities, each mapped into one node of a new graph. The process is repeated until all of the nodes in each small community have been covered. In our scenario, communities represent a set of related amino acid. After the communities have been extracted, we map them into functional regions, i.e., the RBD domain, and analyse the presence of mutations in such communities. The Louvain algorithm has been widely used in network analysis [35, 36] to discover communities. We apply it here to PCNs to find communities overlapping with protein structures and to relate communities with protein functionalities. For each variant, we plot its mutations inside the communities to graphically show where most mutations end up. This allows us to identify a pattern in mutation distributions of the Louvain communities. In the case of a high percentage of mutations belonging to the same community, we can claim that the corresponding mutations share similar effects on protein functionality.

## Structural analysis

We computed TM-scores [37] between pairs of Spike proteins of two different variants by using the US-align (Universal Structural alignment) software [38]. The TM-score quantify the structural similarity between proteins (see the upper part of Fig 1). Thus, the pipeline plots sequence distance and structural distance for Spike variants. We also evaluate contact similarity between PCNs variants. This is defined as the percentage of contacts/non-contacts shared between two PCNs, and should represent a measure of similar behaviours among subnetworks. Finally, for each variant, we computed the acid dissociation constant $pK_a$ for each amino acid of the analysed proteins using the PROPKA3 web server [39]. Given a node, $pK_a$ value is the $-log_{10}K_a$, where $K_a$ is the acid dissociation constant that measures amino acid acidity or basicity. Following the method proposed in [32], the $pKa$ values were used to predict the overall domain charge, also known as EPrbd and EPntd, respectively, for RBD and NTD protein domains. The surface electrostatic potential has been calculated for each variant RBD and NTD by the APBS (Adaptive Poisson-Boltzmann Solver) software with default parameters [40].

## Sequence analysis

We performed sequence alignment using the CLUSTALW software with default parameters [41].

**Implementation.** The methods described above were developed with the help of the JupyterLab environment [42]. The code was written in Python language version 3, plotly and matplotlib libraries (used to draw figures) [43], and PROPKA [39] library to calculate $pK_a$ of the proteins. The PCN-Miner [31] tool computed PCNs, node centralities, and Louvain algorithms to extract communities [30] for the Spike variants. PyMOL library [44] used to read,

visualize and modify PDB files. The CUPSAT software [45] is used to predict changes in protein stability caused by a particular mutation of the Wild Type form. Finally, we used US Align [38] to perform a sequence-independent alignment based on the structural similarity of the variants and to compute the TM score.

## Results

We analysed the sequence and structure of Spike proteins of fourteen selected SARS-CoV-2 variants using PDB files as input. For each PDB file we calculated a PCN. We then evaluate centrality measures (Betweennes, Degree, Eigenvector, Closeness, and Katz) for each PCN. Eigenvector centrality values have been calculated to show how protein structure varies. In particular, we note that when centrality values have little variation among variants, the global form of protein structure does not vary among variants, while local changes occur. All the evaluated centrality measures for variants are reported as boxplots in Fig 3. Boxplots evidence that the changes in Eigenvector Centrality are not significant considering both all nodes and only those corresponding to the mutated residues. Conversely, rewiring of the structure causes more changes in Katz Centrality values than other measures. Further measures confirmed the above results that can be found in Tables 3 and 4.

Fig 2 reports an example of how centrality values are calculated. Fig 3 shows: (a) the comparison of average eigenvector centrality values of the whole S protein, (b) the eigenvector centrality for all the variants, and (c) eigenvector centrality values of the nodes of the RBD domain of all the selected variants. Eigenvector centrality values have been calculated to show changes in structure. We further compared the centrality values of the mutation sites to highlight possible differences on the same variant sites. Our results find no evidence of a homogeneous pattern of change with respect to site: e.g., considering the mutation sites shared by variants, centrality values increase and decrease according to the time of their appearance. Radar plots showing this behaviour are reported in Supplementary materials (see Supporting information Section).

Similarly to [24], the obtained eigenvector centrality evidence protein instability. In particular, the mutations of $Omicron_1$ and Delta variants cause Spike protein instability, according to [14]. Is also woth nothing that for the Wild Type virus, amino acids of the RBD domain

**Table 3. P-values of the t-tests for the comparison of average node centrality values of mutated nodes in the $Omicron_1$ variant only.** EC—Eigenvector Centrality, BC-Betweenness Centrality, KC—Katz Centrality, DC—Degree Centrality, CC—Closeness Centrality.

| Variants couples | p-value EC | p-value BC | p-value KC | p-value DC | p-value CC |
|---|---|---|---|---|---|
| $Omicron_1$ vs Wild Type | 0.655 | 0.353 | 1.148e-10 | 0.181 | 0.091 |
| $Omicron_1$ vs Epsilon | 0.627 | 0.127 | 8.110e-04 | 0.820 | 0.648 |
| $Omicron_1$ vs Zeta | 0.655 | 0.353 | 1.148e-10 | 0.181 | 0.091 |
| $Omicron_1$ vs Beta | 0.1 | 0.00000077 | 1.442-03 | 0.001 | 0.022 |
| $Omicron_1$ vs Alpha | 0.551 | 0.157 | 1.705e-01 | 0.410 | 0.763 |
| $Omicron_1$ vs Delta | 0.22 | 0.00002 | 2.619e-02 | 0.004 | 0.139 |
| $Omicron_1$ vs Kappa | 0.818 | 0.004 | 1.417-01 | 0.380 | 0.934 |
| $Omicron_1$ vs Gamma | 0.674 | 0.011 | 1.053e-04 | 0.888 | 0.923 |
| $Omicron_1$ vs $Iota_1$ | 0.671 | 0.314 | 5.270-11 | 0.242 | 0.099 |
| $Omicron_1$ vs $Iota_2$ | 0.695 | 0.310 | 6.346-03 | 0.663 | 0.825 |
| $Omicron_1$ vs Eta | 0.862 | 0.166 | 3.758e-01 | 0.880 | 0.870 |
| $Omicron_1$ v credo che il s Ihu | 0.008 | 0.105 | 8.761e-10 | 0.014 | 0.088 |
| $Omicron_1$ vs $Omicron_5$ | 0.397 | 0.226 | 6.490e-04 | 0.027 | 0.690 |

**Table 4. P-Values of the t-test for the comparison of average node centrality values of all nodes in the Omicron$_1$ variant.** p-values obtained after correction for multiple tests and with values less than 0.05 have been considered significant. EC—Eigenvector Centrality, BC—Betweenness Centrality, KC—Katz Centrality, DC—Degree Centrality, CC—Closeness Centrality.

| Variants couples | p-value EC | p-value BC | p-value KC | p-value DC | p-value CC |
|---|---|---|---|---|---|
| Omicron$_1$ vs Wild Type | 0.725 | 3.293e-31 | 1.340e-119 | 1.991e-18 | 0.00002 |
| Omicron$_1$ vs Epsilon | 0.925 | 1.097e-02 | 2.378e-45 | 1.323e-03 | 0.0597 |
| Omicron$_1$ vs Zeta | 0.725 | 3.293e-31 | 1.340e-119 | 1.991e-18 | 0.00002 |
| Omicron$_1$ vs Beta | 0.192 | 4.850e-13 | 1.120e-23 | 2.037e-07 | 0.067 |
| Omicron$_1$ vs Alpha | 0.591 | 1.865e-04 | 1.328e-01 | 1.334e-11 | 0.555 |
| Omicron$_1$ vs Delta | 0.027 | 4.552e-12 | 5.673e-23 | 3.599e-08 | 0.050 |
| Omicron$_1$ vs Kappa | 0.609 | 1.067e-05 | 7.881e-19 | 1.423e-07 | 0.266 |
| Omicron$_1$ vs Gamma | 0.916 | 7.393e-01 | 3.139e-57 | 2.852e-04 | 0.773 |
| Omicron$_1$ vs Iota$_1$ | 0.725 | 3.293e-31 | 1.340e-119 | 1.991e-18 | 0.00002 |
| Omicron$_1$ vs Iota$_2$ | 0.930 | 7.954e-07 | 2.450e-23 | 6.653e-04 | 0.405 |
| Omicron$_1$ vs Eta | 0.591 | 1.865e-04 | 1.328e-01 | 1.334e-11 | 0.555 |
| Omicron$_1$ vs Ihu | 0.643 | 2.318e-04 | 2.132e-34 | 4.292e-04 | 0.519 |
| Omicron$_1$ vs Omicron$_5$ | 0.527 | 3.886e-05 | 3.349e-20 | 1.644e-05 | 0.946 |

present the highest eigenvector centrality values. The Omicron$_1$ variant has 15 mutations in the RBD domain, 14 of which can be related to protein instability, whereas the N501Y mutation site is the only Omicron$_1$ mutation in the RBD that increases protein stability. In the Omicron$_1$ PCN, the decreased protein stability caused by mutations in RBDs leads to a decrease in the RBD eigenvector centrality values as reported in Fig 4.

Changes in centrality measures have been compared by using a t-test. A p-value less than 0,05 represent a significant change in average centrality values. Table 3 reports a comparison between Omicron$_1$ and all the other structures considering only mutated sites. Table 4 reports the same comparison for all the residues and Table 5 reports confidence intervals of average centrality measure values.

By using the here proposed pipeline we found a significant change of Katz centrality measures between Omicron$_1$ variant and all the other Spike variants (also considering the Wild Type one). Significant changes in betweenness centrality and degree centrality values have also been reported, where the significance is measured by using t-test analyses. Complete comparisons, code and numerical results can be found in references reported in Supporting information Section.

As reported in Fig 1, after the centrality analysis we analysed the node communities in each PCN were analysed by using the Louvain algorithm, as reported in Fig 5. Figs 6 and 7 provide community detection results. The obtained communities are similar to the functional domain of the protein. In particular Fig 6 reports a comparison of communities of Wild Type, Delta, and Omicron$_1$. Fig 7 reports a comparison between communities in Delta and Omicron$_1$ variant. Table 6 summarises the extracted communities using the Louvain algorithm.

Protein structures were analyzed by means of TM-scores, structural distance, and PCN Contact similarity between Spike variants. Sequence distances between variants were used to construct a phylogenetic tree. Fig 8 reports the comparison of these measures. The figure shows that there is no direct correlation between sequence changes and structural modification. For instance, Iota1, and Zeta variants have dissimilar sequences but highly similar structures.

For each variant, we computed the net charge of RBD and NTD domain. In Fig 9 we show the RBD and NTD net charges of all the selected variants. The values are reported in Table 7.

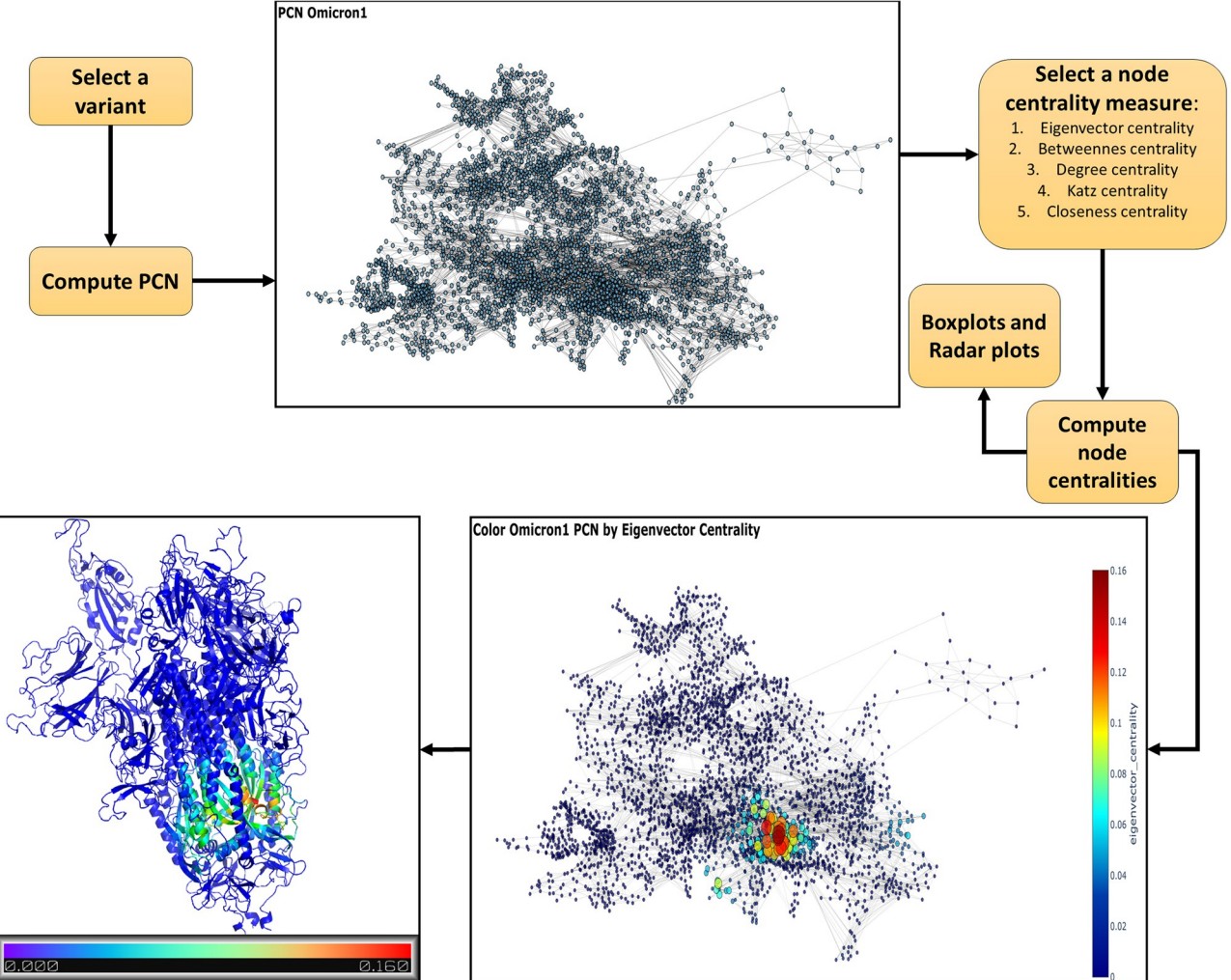

**Fig 2. Centrality analysis pipeline: Starting with a Spike protein variant we compute the PCN.** Then, we calculate centrality values for each node and finally map values on the protein structure with different colors.

## Discussion

Results in Tables 3 and 4 show that changes in centrality values could signify protein instability. Considering $Omicron_1$ and Delta variants in particular (see Fig 4) the potential instability is due to the following specific mutations. E.g., the $Omicron_1$ variant had multiple mutations in the RBD domain, most of which associated with protein instability. The N501Y mutation is the only one this increased stability. Consequently, the $Omicron_1$ PCN presents an eigenvector centrality value that decreases for the RBD domain.

The significance of changes in centrality measures was assessed by t-test analyses, comparing average values across nodes and nodes corresponding to mutations. Variations in centrality measures, including Katz centrality, between $Omicron_1$ and other variants (including the Wild Type) were found to be significant. Changes in betweenness centrality and degree centrality values were also reported. Community detection analysis using the Louvain algorithm identified communities within PCNs, which often correspond to functional domains of the Spike protein. Mutations occurring within the same community appeared to have similar

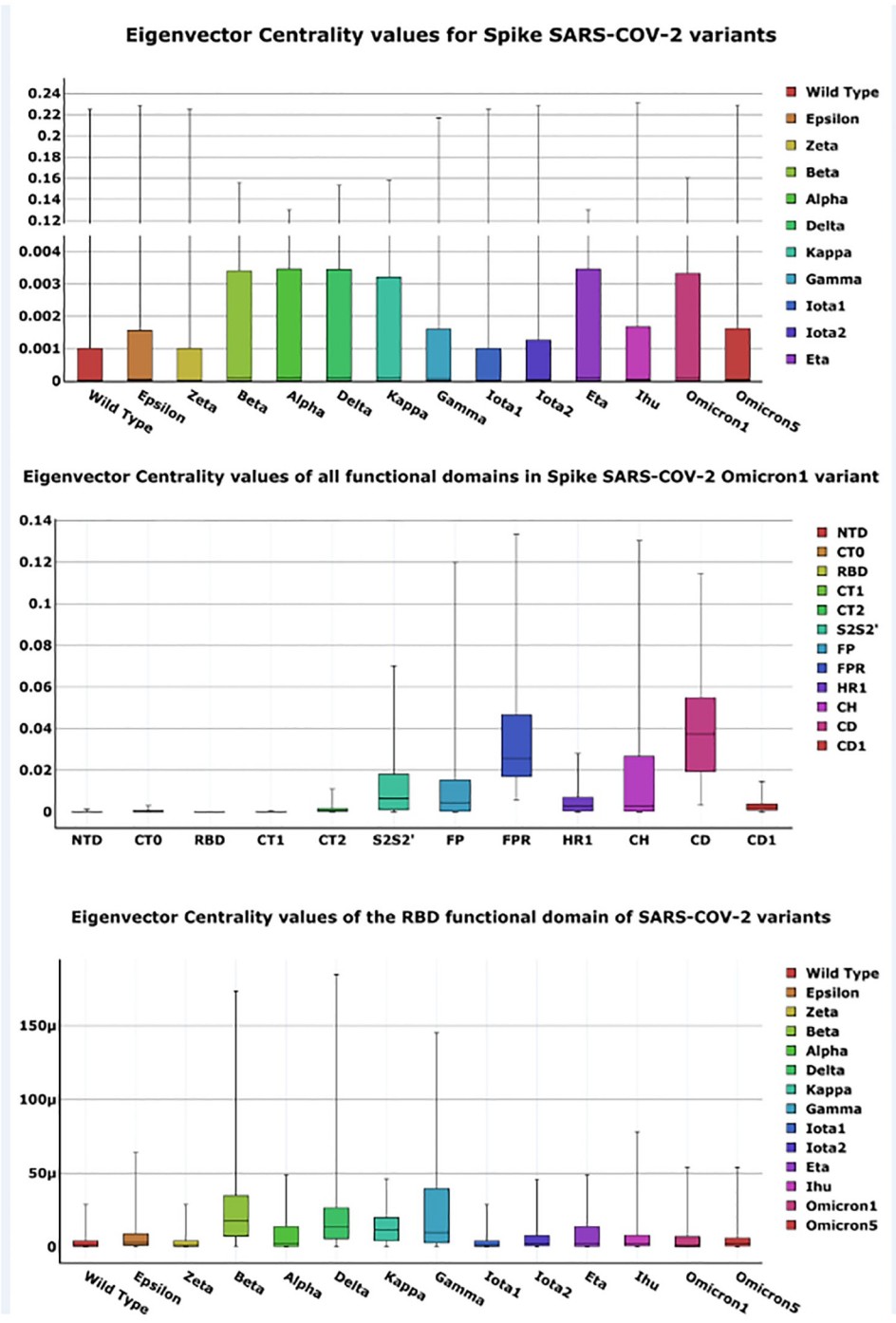

**Fig 3. Node eigenvector centrality boxplots.** From the upper part of the figure: (a) average centrality values of all the nodes of the S proteins of the selected variants as boxplots; (b) the eigenvector centrality values for nodes of the functional domain (Omicron$_1$ variant); (c) eigenvector centrality values of the nodes of the RBD domain for all the selected variants.

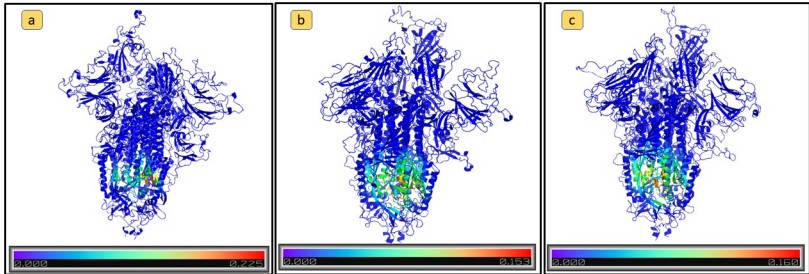

**Fig 4. Amino acids eigenvector centrality values mapped on the protein structure of the following Spike variants: a) Omicron$_1$; b) Wild Type; c) Delta.** Eigenvector centrality values are represented by a color-based scale from blue (lower values) to red (higher values). The decrease of eigenvector centrality of RBD domain of the Omicron$_1$ variants is shown by the presence of a larger number of blue colored nodes.

effects on protein function. The resulting communities and their mapping onto protein structures are reported in Figs 5–7. Omicron$_1$ and Omicron$_5$ RBD mutations belong to the same community. This may imply a similar impact of the mutations on the Spike protein function in the two variants. Conversely, the Delta variant presents mutation in different communities, thus suggesting a remarkable difference between Omicron$_1$ and Delta. Protein structure analysis as performed by measuring TM-scores, structural distances, and PCN Contact similarity between Spike variants. Sequence distances were mapped into a phylogenetic tree. Fig 8 illustrates the comparison of these measures. In particular parts a, b and c of the figure represent, respectively: (a) the heatmap of the structural distance; (b) the heatmap for PCNs similarity and (c) phylogenetic tree representing distances among sequences. As reported in Fig 8, there is no correlation between sequence and structure similarity [28].

The acid dissociation constant (pKa) used to predict the net charge showed that protein Spike acquires a positive charge with time as shown in Fig 9.

Valuable insights into the centrality and community structure of Spike protein variants were gained, suggesting potential protein instability, identifying mutation clusters, and assessing structural characteristics. Results contribute to understanding the behaviours and potential implications of different SARS-CoV-2 variants. Moreover, they have several important

**Table 5. Confidence intervals (CI) of average centrality values (from left to right: Eigenvector, Betweennes, Katz, Degree, and Closeness).**

| Variant | CI EC | CI BC | CI KC | CI DC | CI CC |
|---|---|---|---|---|---|
| Epsilon | [0.00065, 0.00384] | [-0.00025, 0.001] | [0.00669, 0.02141] | [0.00146, 0.00311] | [0.05569, 0.08355] |
| Zeta | [-0.00023, 0.0041] | [-6e-05, 0.00045] | [0.00272, 0.01846] | [0.00093, 0.00294] | [0.05656, 0.09342] |
| Beta | [0.00144, 0.00372] | [-0.00054, 0.0086] | [0.00855, 0.01676] | [0.00232, 0.00307] | [0.06988, 0.08025] |
| Alpha | [-3e-05, 0.00855] | [-0.0001, 0.00722] | [0.00663, 0.01195] | [0.002, 0.00244] | [0.0715, 0.08142] |
| Delta | [0.00044, 0.0052] | [-0.00041, 0.00745] | [0.00808, 0.01489] | [0.00244, 0.00339] | [0.066, 0.07829] |
| Kappa | [0.00038, 0.00629] | [5e-05, 0.00607] | [0.00366, 0.01009] | [0.00141, 0.00232] | [0.0609, 0.07708] |
| Gamma | [0.00138, 0.005] | [-0.00088, 0.00858] | [0.01118, 0.01762] | [0.00177, 0.0023] | [0.0668, 0.07842] |
| Iota1 | [0.00065, 0.00306] | [-2e-05, 0.00022] | [0.0084, 0.01967] | [0.00158, 0.00289] | [0.0597, 0.07749] |
| Iota2 | [0.00054, 0.0035] | [-4e-05, 0.00032] | [0.00736, 0.01741] | [0.00168, 0.00286] | [0.0588, 0.07832] |
| Eta | [0.00094, 0.00338] | [-0.00134, 0.01647] | [0.00578, 0.01854] | [0.00188, 0.00246] | [0.06805, 0.07824] |
| Ihu | [0.00195, 0.00552] | [-3e-05, 0.00315] | [0.01035, 0.01484] | [0.00194, 0.00251] | [0.06905, 0.07511] |
| Omicron1 | [0.00183, 0.00367] | [0.00025, 0.0008] | [0.00631, 0.00784] | [0.00185, 0.00219] | [0.06907, 0.07517] |
| Omicron5 | [0.00152, 0.003] | [7e-05, 0.00046] | [0.00759, 0.01001] | [0.00184, 0.00219] | [0.06805, 0.07375] |

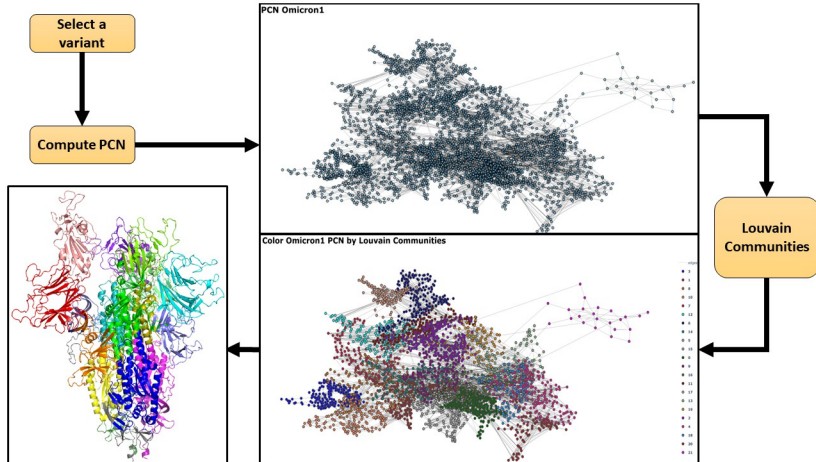

**Fig 5. Starting with a Spike protein variant, for example the Omicron$_1$ variant, we use the corresponding PDB file to compute a PCN.** Then, communities are identified and plotted by applying the Louvain community detection algorithm on the PCN. Finally, communities are mapped on the protein structure to relate communities and functional domains of the protein.

biological implications. Topological analysis of protein contact networks can provide insight into the stability and function of proteins. The results regarding changes in centrality measures indicate protein instability, with the Omicron$_1$ variant showing decreased RBD eigenvector centrality due to mutations associated with protein instability. This information may be further analysed to shed light on the effects of viral mutations on transmissibility and changes in disease severity. This could be considered the first step towards the prediction of novel mutations as well as support for providing therapies and vaccines that target specific regions of the virus. Moreover, the community detection analysis revealed that mutations falling in the same community often have similar effects on protein structure and hence function. This may help to predict the effects of mutations on the virus's behaviour.

We have shown that many changes occurred after the beginning of the vaccination campaign. These changes include modifications of protein structure as evidenced through TM-scores, structural distance, and PCN contact similarity. This may help to explain how virus

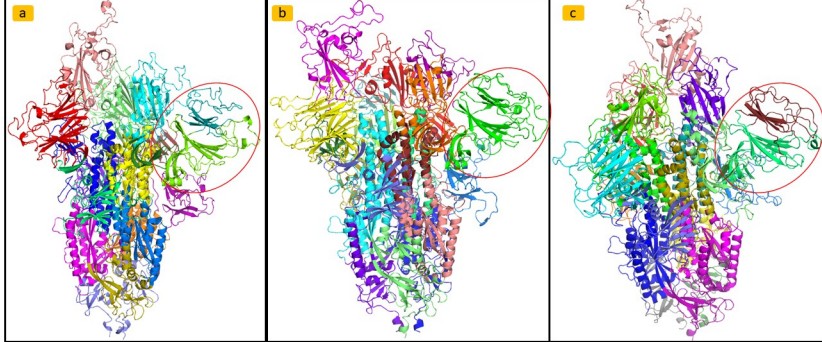

**Fig 6. Community detection analysis comparing Spike of the Wild Type, Delta, and Omicron$_1$.** Communities mapped directly on the protein structure of a) Wild Type, b) Delta, and c) Omicron$_1$ variant to visualize functional domains predicted by the Louvain algorithm.

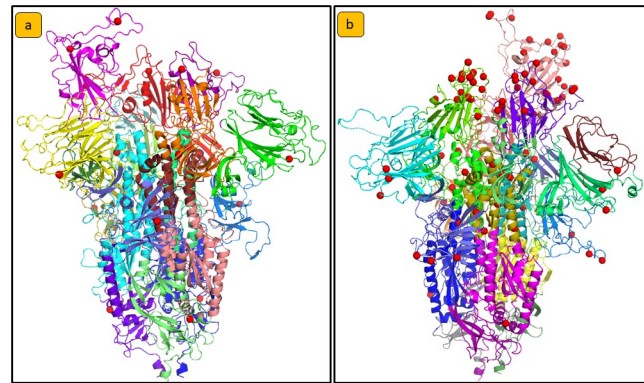

**Fig 7. Community detection analysis comparison between a) Delta and b) Omicron$_1$.** Visualization of the mutations and the communities on the protein structure. Mutated residues are displayed as red spheres. In Omicron$_1$, and Omicron$_5$, the mutations seem to fall inside certain communities with the same function in the Spike Protein. We found that Omicron$_1$ has more than ten mutations that fall inside the same community.

mutations affect viral transmission, replication, or immune evasion. This information can be used to guide the design of drugs and vaccines that target specific regions of the virus and can help to predict the spread and evolution of new variants. We explore patterns of changes in a temporal dimension and compare the cumulative distribution of vaccination with the characteristics of the variant. Although we cannot infer any causality regarding vaccination driving the evolution, we should note that the presence of vaccinations in a timeline is located in the middle of the first variants of SARS-CoV-2 and Omicron. Considering also the clinical characteristics of Omicron in terms of vaccine escape and neutralization of immune response, we can assume that the effect of all Omicron changes may be related to the structural changes also revealed by the above-reported measures.

**Table 6. Summary of communities and their mutations.** As some mutations do not belong to any community, we report them here as belonging to a virtual community referred to as (UnModelled).

| Variant | No. of communities | Total No. of mutations | Mutations for each Community (community No: No of mutations) |
|---|---|---|---|
| Wild Type | 22 | 0 | |
| Epsilon | 22 | 4 | UM: 1, Community 0: 1, Community 4: 1, Community 16: 1 |
| Zeta | 19 | 3 | Community 9: 1, Community 8: 1, UM: 1 |
| Beta | 21 | 8 | Community 4: 3, Community 15: 3, Community 11: 1, Community 7: 1 |
| Alpha | 22 | 7 | Community 10: 1, Community 11: 1, Community 12: 2, Community 18: 2, Community 6: 1 |
| Delta | 22 | 6 | Community 1: 3, Community 15: 2, Community 0: 1 |
| Kappa | 23 | 5 | Community 1: 2, Community 15: 2, Community 5: 1 |
| Gamma | 19 | 8 | Community 3: 2, Community 6: 2, Community 16: 1, Community 10: 2, Community 4: 1 |
| Iota1 | 21 | 5 | UM: 1, Community 2: 2, Community 15: 1, Community 9: 1 |
| Iota2 | 21 | 5 | UM: 1, Community 0: 2, Community 15: 1, Community 8: 1 |
| Eta | 24 | 6 | Community 19: 2, Community 1: 1, Community 12: 2, Community 10: 1 |
| Ihu | 24 | 13 | Community 15: 1, Community 1: 4, Community 7: 3, Community 12: 2, Community 0: 2, UM: 1 |
| Omicron$_1$ | 20 | 30 | Community 2: 4, Community 5: 15, Community 7: 1, Community 9: 4, Community 13: 4, Community 0: 2 |
| Omicron$_5$ | 19 | 27 | Community 2: 10, Community 7: 8, Community 17: 1, Community 3: 4, Community 13: 2, Community 0: 2 |

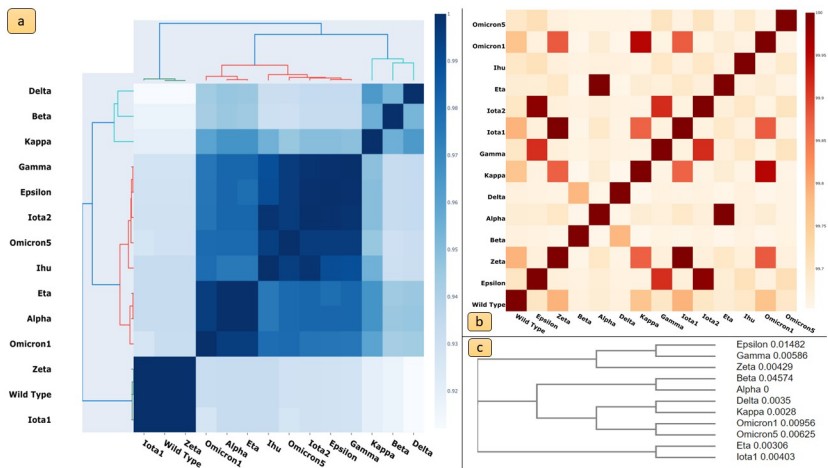

**Fig 8.** Structural, Sequence and Network similarity results: a) Heatmap representing the structural similarity of variants computed by TM-scores; b) Heatmap representing the similarity between PCNs of variants; c) sequence similarity represented by a Phylogenetic tree. The figure evidences no direct correlation between sequence changes and structural modification. For instance, Iota1, and Zeta variants have dissimilar sequences but very similar structures.

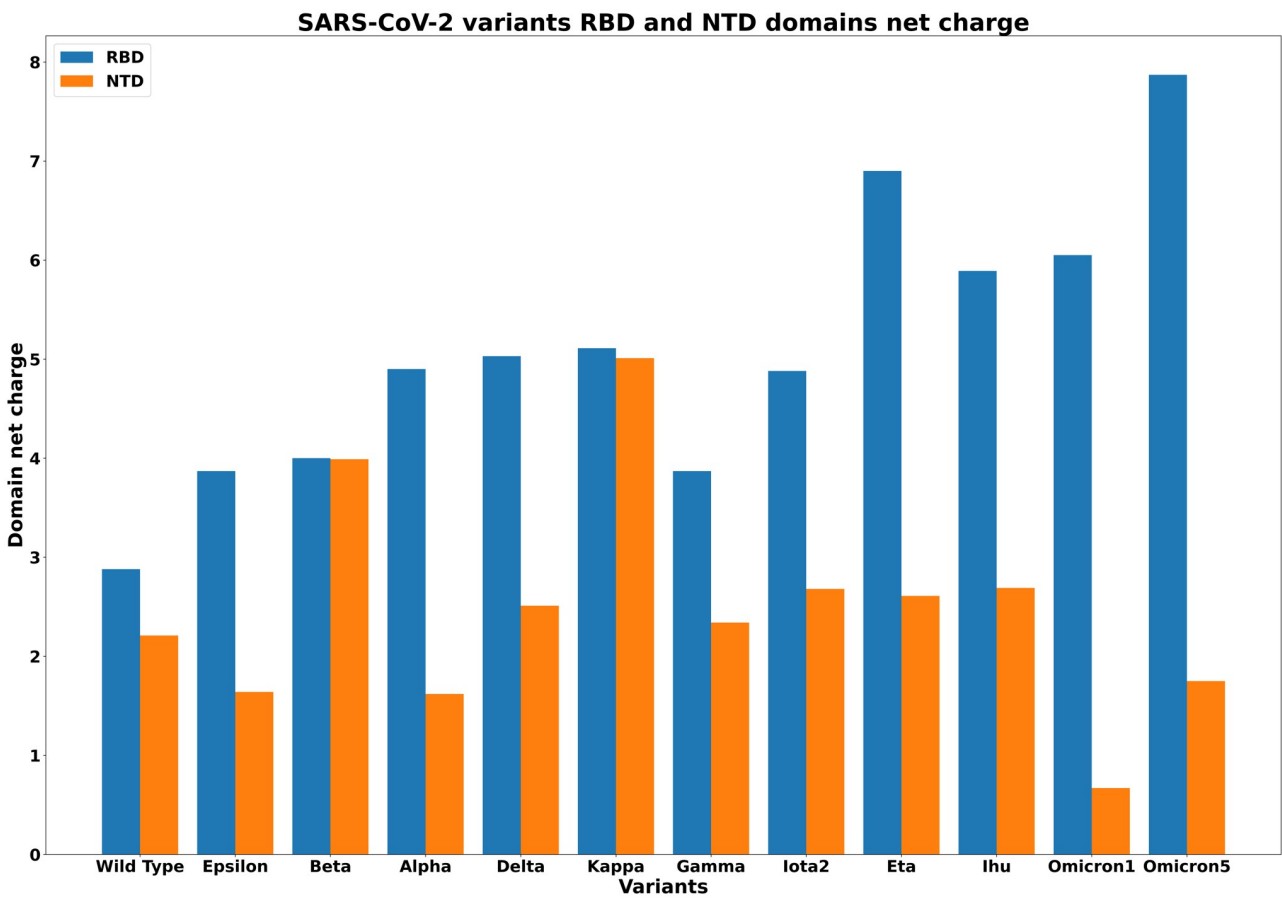

**Fig 9. RBD and NTD domain net charges for all the selected variants.**

**Table 7. Net charge of the Spike protein variants in RBD and NTD domains.** With $e$ the elementary charge constant equals to $1.602 * 10^{-19}C$.

| Variants | Domain | Folded | Unfolded |
|---|---|---|---|
| SPWT | NTD | 2.21e | 2.83e |
| SPWT | RBD | 2.88e | 2.92e |
| Epsilon | NTD | 1.64e | 1.91e |
| Epsilon | RBD | 3.87e | 3.92e |
| Beta | NTD | 3.99e | 5.37e |
| Beta | RBD | 4.0e | 4.11e |
| Alfa | NTD | 1.62e | 1.92e |
| Alfa | RBD | 4.9e | 4.91e |
| Delta | NTD | 2.51e | 3.37e |
| Delta | RBD | 5.03e | 5.14e |
| Kappa | NTD | 5.01e | 5.34e |
| Kappa | RBD | 5.11e | 5.15e |
| Gamma | NTD | 2.34e | 2.83e |
| Gamma | RBD | 3.87e | 3.91e |
| Iota | NTD | 2.68e | 2.92e |
| Iota | RBD | 4.88e | 4.91e |
| Eta | NTD | 2.61e | 2.92e |
| Eta | RBD | 6.9e | 6.91e |
| Ihu | NTD | 2.69e | 2.93e |
| Ihu | RBD | 5.89e | 5.92e |
| Omicron1 | NTD | 0.67e | 0.92e |
| Omicron1 | RBD | 6.05e | 6.15e |
| Omicron5 | NTD | 1.75e | 1.83e |
| Omicron5 | RBD | 7.87e | 7.91e |

Finally, our results show that Omicron variants clearly differ from the others. All the considered parameters confirm that there is a remarkable change in centrality values. In particular, the difference in terms of network invariants among Alpha, Beta, Delta, and Omicron has been studied also in [28]. The presented results here can be considered as a further extension of our previous in two ways. First, we confirm that the difference is not limited to network invariants, but also other protein structural measures, i.e. polarity and TM-Score confirm that Omicron is radically different from the others. We also note an increase of net charge of RBD domain over time [32] which may explain the increased transmissibility through a better binding affinity with the ACE2 receptor.

Finally, as we also show in Fig 10, Omicron$_1$ variants appeared after the start of the vaccination campaign. Even though this does not imply any causal relation [46, 47], this relation may require additional studies to shed light on the relation between the vaccination campaign and the evolution of the SARS-CoV-2 virus.

## Study limitations

While the study provides valuable insights into the centrality and community structure of Spike protein variants, there are some limitations to consider:

1. **Data Availability:** The study is based on the availability of protein structure encoded into PDB files, which might have limitations in terms of availability, completeness, and accuracy.

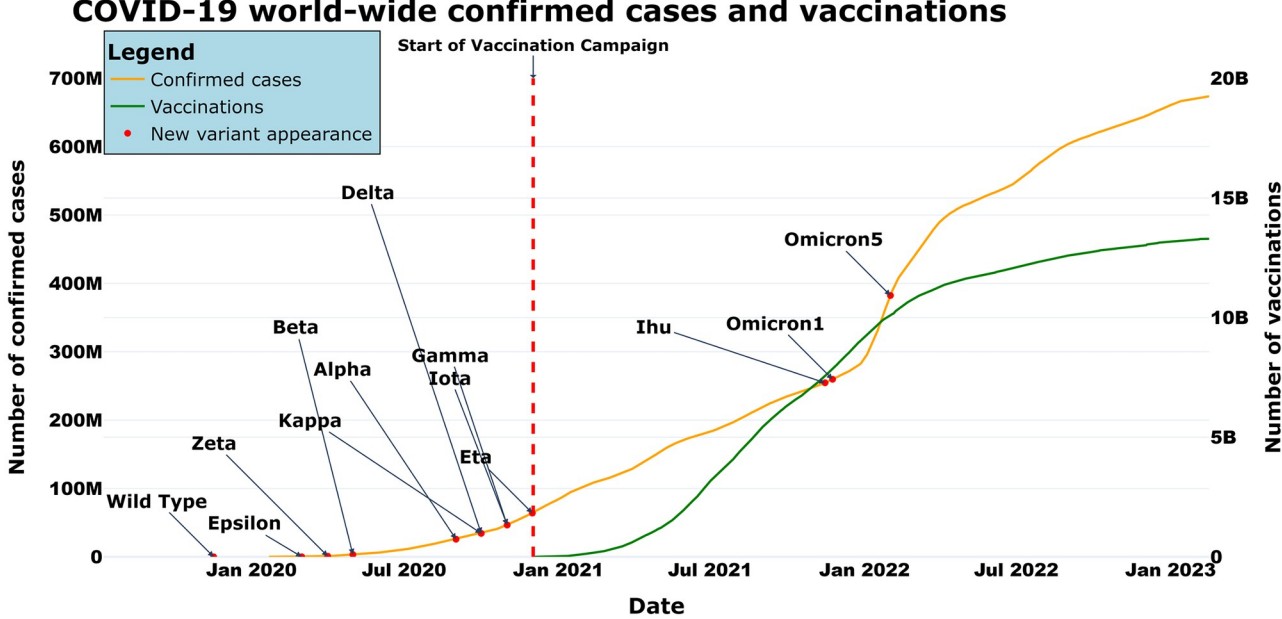

**Fig 10. COVID-19 worldwide number of vaccinations and confirmed cases studied.** Variants are reported as points on their first detection date.

2. **Simplified Representation:** The analysis considers only protein structures and network structures while other factors such as protein dynamics, ligand interactions, and post-translational modifications might provide more insights.

3. **Lack of Experimental Validation:** As the findings are based on in-silico, experimental validation is needed to verify structural changes.

4. **Limited Sample Size:** As the analysis focuses on a specific set of Spike protein variants, the inclusion of a larger set of variants could corroborate the results.

Further research addressing these limitations could contribute to a fuller understanding of the implications of Spike protein variants on protein structure and function.

## Conclusion

In this work we presented a pipeline for analysing mutations of the SARS-CoV-2 genome, focusing on the impact of the mutations on Spike protein. From a timeline perspective, we observed that the Omicron variant presents significant changes with respect to the previous ones and Omicron appeared in parallel with the worldwide vaccination campaign. Omicron's structure presents many changes in the RBD domain and many mutations fall within the same structural region. A final remark, therefore we would argue that further studies should focus on the possible relationship between the vaccination campaign and the of Omicron appearance.

## Supporting information

**S1 Fig. Clustermap of communities of Spike protein.** Clustermap plot for a) Wild Type, b) Delta, and c) Omicron$_1$ variant. In a clustermap plot, communities are mapped in a matrix and visualized by means of heatmap.
(TIF)

**S1 File.**
(DOCX)

**S1 Graphical abstract.**
(PNG)

## Author Contributions

**Conceptualization:** Ugo Lomoio, Barbara Puccio, Giuseppe Tradigo, Pietro Hiram Guzzi, Pierangelo Veltri.

**Data curation:** Ugo Lomoio, Barbara Puccio, Pietro Hiram Guzzi, Pierangelo Veltri.

**Formal analysis:** Barbara Puccio, Pietro Hiram Guzzi.

**Methodology:** Pierangelo Veltri.

**Project administration:** Pierangelo Veltri.

**Software:** Ugo Lomoio, Barbara Puccio.

**Supervision:** Ugo Lomoio, Giuseppe Tradigo, Pietro Hiram Guzzi, Pierangelo Veltri.

**Validation:** Pietro Hiram Guzzi.

**Writing – original draft:** Ugo Lomoio, Barbara Puccio, Giuseppe Tradigo, Pietro Hiram Guzzi, Pierangelo Veltri.

**Writing – review & editing:** Giuseppe Tradigo, Pietro Hiram Guzzi, Pierangelo Veltri.

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
