## [Decision Letter · Decision Letter 0]

4 May 2023

PONE-D-23-06693SARS-CoV-2 protein structure and sequence mutations: evolutionary analysis and effects on virus variantsSARS-CoV-2 protein structure and sequence mutations:PLOS ONE

Dear Dr. Guzzi,

Thank you for submitting your manuscript to PLOS ONE. After careful consideration, we feel that it has merit but does not fully meet PLOS ONE’s publication criteria as it currently stands. Therefore, we invite you to submit a revised version of the manuscript that addresses the points raised during the review process.

We look forward to receiving your revised manuscript.

Kind regards,

Nagarajan Raju

Academic Editor

PLOS ONE

Journal Requirements:

4. Please upload a new copy of Figure 11b as the detail is not clear. Please follow the link for more information: " ext-link-type="uri" xlink:type="simple">https://blogs.plos.org/plos/2019/06/looking-good-tips-for-creating-your-plos-figures-graphics/"
" ext-link-type="uri" xlink:type="simple">https://blogs.plos.org/plos/2019/06/looking-good-tips-for-creating-your-plos-figures-graphics/"

Additional Editor Comments:

I suggest authors to go through comments from all the reviewers and include the responses in the revised version

Reviewers' comments:

Reviewer's Responses to Questions

**Comments to the Author**

1. Is the manuscript technically sound, and do the data support the conclusions?

Reviewer #1: Partly

Reviewer #2: Yes

Reviewer #3: Partly

Reviewer #4: No

Reviewer #5: Yes

Reviewer #6: Yes

2. Has the statistical analysis been performed appropriately and rigorously? 

Reviewer #1: N/A

Reviewer #2: Yes

Reviewer #3: I Don't Know

Reviewer #4: No

Reviewer #5: Yes

Reviewer #6: I Don't Know

3. Have the authors made all data underlying the findings in their manuscript fully available?

Reviewer #1: Yes

Reviewer #2: Yes

Reviewer #3: Yes

Reviewer #4: No

Reviewer #5: Yes

Reviewer #6: Yes

4. Is the manuscript presented in an intelligible fashion and written in standard English?

Reviewer #1: No

Reviewer #2: Yes

Reviewer #3: Yes

Reviewer #4: No

Reviewer #5: Yes

Reviewer #6: Yes

5. Review Comments to the Author

Reviewer #1: The manuscript is easy to understand and the technical implementation is sound. The manuscript seems to present a pipeline. The SARS-CoV-2-2 protein appears to be an example to present the possibilities of the pipeline and the number of 15 plots and four tables. I miss a biological question or motivation. The claims in the introduction are vague. Network properties such as the Betweenness centrality are textbook knowledge that may not need an introduction. The manuscript should, however, introduce all quantities used in the explanation, e.g., what is the matrix A in the eigenvector centrality? The resolution of some figures, e.g., figure 2, is too low to see anything. The message of the majority of the figures remains unclear. There are some minor typos in the manuscript.

Reviewer #2: Dear authors,

Your manuscript your manuscript entitled "SARS-CoV-2 protein structure and sequence mutations: evolutionary analysis and effects on virus variants" is interesting. However, I have a few suggestions that needed to be addressed-

1. The abstract needs to be re-written as it feels like a undergraduate student have written it.

2. Read and cite the following paper that has addressed the same issue. https://doi.org/10.1007/s10753-022-01734-w

3. Make a better graphical abstract that can decipher the work better for the complete study and connect each formula in theoretical way to make it more correlative.

4. Make a separate discussion heading and discuss your work against state of the art work and above provided paper.

I wish to see the above changes to re-review it critically.

Reviewer #3: The work is on the effect of mutations of the structure of Spike protein in SARS-CoV-2. The work described here is heavily dependent upon protein contact networks (PCN). The works seems okay overall though. Therefore, I am recommending it for publication.

Reviewer #4: Comments to the author

1. The abstract does not provide enough context about the research, such as the specific research question, hypothesis and findings. It only provides general information about the relationship between protein sequence, structure, and function, and the use of Protein Contact Networks (PCNs) to investigate protein structures. Authors are suggested to provide a clear and concise summary of the main findings and implications of the analysis.

2. The explanation of the Louvain community detection analysis could be more detailed to provide a better understanding of the approach.

3. While the PKa values are reported for three variants, it is unclear why only these variants were selected and how these values were calculated.

4. Authors should mention all the software and tools those are used to perform the analysis.

5. The study results are presented without proper context or background information, making it challenging for the reader to understand the subject matter and the findings. The significance of the results is not adequately discussed, and the authors do not provide any recommendations or conclusions based on their findings.

6. Although the authors mention that they performed t-test analyses to determine the significance of their results, they do not provide any details about the statistical tests performed, such as the p-values or confidence intervals.

7. It is unclear how the results of the acid dissociation constant (PKa) analysis relate to the overall findings of the study. A more detailed explanation of the implications of this analysis would be helpful.

8. While the authors identify significant changes in centrality measures between Omicron1 and other Spike variants, they do not explain the biological significance or potential implications of these changes.

9. Authors unnecessarily make many figures. Several figures could be merged into one. Also, the text in the figures should be clear.

10. There is no discussion of the limitations of the study or potential sources of bias.

Reviewer #5: In this paper, the authors are proposed “SARS-CoV-2 protein structure and sequence mutations: evolutionary analysis and effects on virus variantsSARS-CoV-2 protein structure and sequence mutations”

The strengths of the paper are that it is well structured, the description of the related work is well done and that results are extensively compared to results of the similar research.

Minor revisions:

1. Authors should draw a graphical abstract of the proposed approach

2. Authors should justify the proposed approach and compare your approach with existing algorithms.

3. Proofread the entire manuscript.

Reviewer #6: The authors shared a rather good paper. The analysis outcomes presented are in alignment with what is already known from the biology of the virus. However, some extra work is needed to improve the English language of the paper. I see some verbs explaining the methods are written in the present tense.

6. PLOS authors have the option to publish the peer review history of their article (what does this mean?). If published, this will include your full peer review and any attached files.

Reviewer #1: **Yes: **Jörg Ackermann

Reviewer #2: **Yes: **Shaban Ahmad

Reviewer #3: **Yes: **Ishtiaque Ahammad

Reviewer #4: No

Reviewer #5: No

Reviewer #6: **Yes: **Rehab Ahmed

---

## [Author Response · Author response to Decision Letter 0]

10 Jun 2023

Reviewer #1: The manuscript is easy to understand and the technical implementation is sound. The manuscript seems to present a pipeline. The SARS-CoV-2-2 protein appears to be an example to present the possibilities of the pipeline and the number of 15 plots and four tables. I miss a biological question or motivation. The claims in the introduction are vague.

Answer: We focus on the correlation between sequence and structure evolution and figure out how sequence updates can be related to phenotypic variations in structures. Moreover, the study regards Spike proteins and Omega variation for Sars-Covid-19. We added such a reference in the introduction and updated claims adding references to figures and results and removed some, for greater concision.

Network properties such as the Betweenness centrality are textbook knowledge that may not need an introduction.

Answer: We thank the reviewer and agree that the definition of measures can be found in textbooks. However, some reviewers from other journals asked for formal introduction. We removed them from the text and included them in a table.

The manuscript should, however, introduce all quantities used in the explanation, e.g., what is the matrix A in the eigenvector centrality?

Answer: We added missing definitions (e.g., adjacency matrix A, now included in the Table for description).

The resolution of some figures, e.g., figure 2, is too low to see anything.

Answer: We produced novel figures improving quality and readability. We have revised all figures, removing those that, in line with the reviewer’s observations, lack clarity.

The message of the majority of the figures remains unclear.

There are some minor typos in the manuscript.

Answer: We thank the reviewer. We corrected the manuscript and typos.

Reviewer #2: Dear authors,

 Your manuscript entitled "SARS-CoV-2 protein structure and sequence mutations: evolutionary analysis and effects on virus variants" is interesting. However, I have a few suggestions that needed to be addressed-

 1. The abstract needs to be re-written as it feels like a undergraduate student have written it.

Answer: we thank the reviewer for the observation. We have rewritten the abstract in order to link it more closely with background and motivations (first paragraph) and model proposal (contributions) and results (applications) as described in the second paragraph.

The abstract has been thus rewritten, hopefully more fluently.

 2. Read and cite the following paper that has addressed the same issue. https://doi.org/10.1007/s10753-022-01734-w

Answer: We thank the reviewer, the paper has been cited, even if we believe that our model goes somewhat further than the cited proposal.

 3. Make a better graphical abstract that can decipher the work better for the complete study and connect each formula in a theoretical way to make it more correlative.

Answer: We provided a graphical abstract to improve readability, and related formulas throughout the text.

The graphical abstract now has the following caption: “Graphical abstract reporting the contribution: Sequence and structure of variants are used as input, PCNs and pKa are evaluated Louvain and centrality values are evaluated on PCNs. Sets of communities of PCNs, centrality values and pka values for mutations are used to characterise how protein (sequence and structure) mutations are related in time with vaccinations and new variants.”

 4. Make a separate discussion heading and discuss your work against state of the art work and above provided paper.

Answer: thank you for the suggestion. In line with the submission format we have added a Discussion Section.

 I wish to see the above changes to re-review it critically.

Reviewer #3: The work is on the effect of mutations of the structure of Spike protein in SARS-CoV-2. The work described here is heavily dependent upon protein contact networks (PCN). The works seems okay overall though. Therefore, I am recommending it for publication.

We thank the reviewer for the helpful comments.

Reviewer #4: Comments to the author

 1. The abstract does not provide enough context about the research, such as the specific research question, hypothesis and findings. It only provides general information about the relationship between protein sequence, structure, and function, and the use of Protein Contact Networks (PCNs) to investigate protein structures.

Answer: We have restructured the abstract in line with the above suggestion. In particular, we added the specific research questions and findings. We also highlighted research questions about the Investigation of connections and relationships between sequence modifications and structural changes in Spike Protein. We focus on the relations between sequence evolution in S protein sequence, structure and thus functions with virus transmissibility and vaccination effects. Studies of S protein generated large data sets involving sequences and structures.

We also enriched the discussion by focusing on how (i) node centrality and (ii) community extraction analysis can be considered to relate protein stability and functionality with sequence mutations. Starting from such metrics, we compare structural evolution to sequence changes, and study mutations from a temporal perspective focusing on virus variants. We apply our model to the Omicron variant highlighting a timeline correlation between the Omicron variant and the vaccination campaign.

Authors are suggested to provide a clear and concise summary of the main findings and implications of the analysis.

Answer: We added in the introduction a list of the main findings and results.

 2. The explanation of the Louvain community detection analysis could be more detailed to provide a better understanding of the approach.

Answer: we added a more detailed explanation of the Louvain community detection analysis in the Methods section.

 3. While the PKa values are reported for three variants, it is unclear why only these variants were selected and how these values were calculated.

Answer: We thank the reviewer for this observation. In the submitted version we focused only on Variants of Concern. We run the experiments considering results for all the known variants (i.e. not only VOC). We used the PROPKA3 web server (https://www.ddl.unimi.it/vegaol/propka_about.htm), to predict pKa values for each amino acid starting from the PDB structure. A similar approach is used in Pascarella et al. J Infect. 2022 May; 84(5): e62–e63. We have also added a detailed discussion of our results.

4. Authors should mention all the software and tools those are used to perform the analysis.

Answer: We checked the paper carefully to ensure that all the software used are mentioned.

5. The study results are presented without proper context or background information, making it challenging for the reader to understand the subject matter and the findings.

Answer: we provided more information about the context in both the abstract and introduction.

The significance of the results is not adequately discussed, and the authors do not provide any recommendations or conclusions based on their findings.

Answer: we have taken this observation- also made by other reviewers- on board and added a Discussion Section, providing additional information and considerations on the obtained results.

 6. Although the authors mention that they performed t-test analyses to determine the significance of their results, they do not provide any details about the statistical tests performed, such as the p-values or confidence intervals.

Answer: We added the requested details and updated Tables for confidence intervals accordingly, while pvalues are reported in the Results Section.

7. It is unclear how the results of the acid dissociation constant (PKa) analysis relate to the overall findings of the study. A more detailed explanation of the implications of this analysis would be helpful.

Answer: We apologise for the lack of clarity. First, we would like to point out that the analysis of the overall charge of the Spike Protein impacts the capability of binding ACE2 and, hence transmissibility. As also shown in Pascarella et al (J Infect. 2022 May; 84(5): e62–e63.) the Pka can be used to predict the overall domain charge, so changes in PkA impact the domain charge and the virus transmissibility. We calculate the changes in the overall charge for the VOC showing that all the variants present a positive change in the net variants. As previously noted in Math et al https://www.ncbi.nlm.nih.gov/pmc/articles/PMC9158474/., similar analysis showed that mutation of the Delta variant made the variants more alkaline, improving their structural robustness. Here we show that Omicron presents a higher increase in net charge. Results confirm that there is a common change in all the variants that give the virus greater transmissibility since Spike has become more positive, while the ACE2 receptor has a negative charge.

8. While the authors identify significant changes in centrality measures between Omicron1 and other Spike variants, they do not explain the biological significance or potential implications of these changes.

Answer: We added a Discussion Section highlighting the biological significance and implications also in terms of applications.

9. Authors unnecessarily make many figures. Several figures could be merged into one. Also, the text in the figures should be clear.

Answer: We apologise for any lack of clarity. To address this weakness, we have restructured all figures, merging some and removing others, as suggested. We also moved figures that were hard to read in paper format into supplementary materials, in the github repository with code available.

10. There is no discussion of the limitations of the study or potential sources of bias.

Answer: thank you for the observation. To improve the paper we have added a Section that follows up the discussion and reports study limitations.

Reviewer #5: In this paper, the authors are proposed “SARS-CoV-2 protein structure and sequence mutations: evolutionary analysis and effects on virus variantsSARS-CoV-2 protein structure and sequence mutations”

 The strengths of the paper are that it is well structured, the description of the related work is well done and that results are extensively compared to results of the similar research.

 Minor revisions:

 1. Authors should draw a graphical abstract of the proposed approach

Answer: We provided a graphical abstract.

2. Authors should justify the proposed approach and compare your approach with existing algorithms.

Answer: We have now presented such a comparison in the introduction.

3. Proofread the entire manuscript

Answer: We have proofread the manuscript with the help of a native English speaker.

Reviewer #6: The authors shared a rather good paper. The analysis outcomes presented are in alignment with what is already known from the biology of the virus. However, some extra work is needed to improve the English language of the paper. I see some verbs explaining the methods are written in the present tense.

Answer: We thank the reviewer. We carefully checked the paper with the help of a native English speaker.

---

## [Decision Letter · Decision Letter 1]

5 Jul 2023

SARS-CoV-2 protein structure and sequence mutations: evolutionary analysis and effects on virus variantsSARS-CoV-2 protein structure and sequence mutations:

PONE-D-23-06693R1

Dear Dr. Guzzi,

We’re pleased to inform you that your manuscript has been judged scientifically suitable for publication and will be formally accepted for publication once it meets all outstanding technical requirements.

Kind regards,

Nagarajan Raju

Academic Editor

PLOS ONE

Additional Editor Comments (optional):

Reviewers' comments:

Reviewer's Responses to Questions

**Comments to the Author**

1. If the authors have adequately addressed your comments raised in a previous round of review and you feel that this manuscript is now acceptable for publication, you may indicate that here to bypass the “Comments to the Author” section, enter your conflict of interest statement in the “Confidential to Editor” section, and submit your "Accept" recommendation.

Reviewer #2: All comments have been addressed

Reviewer #3: All comments have been addressed

Reviewer #5: All comments have been addressed

2. Is the manuscript technically sound, and do the data support the conclusions?

Reviewer #2: Yes

Reviewer #3: Yes

Reviewer #5: Yes

3. Has the statistical analysis been performed appropriately and rigorously? 

Reviewer #2: N/A

Reviewer #3: N/A

Reviewer #5: Yes

4. Have the authors made all data underlying the findings in their manuscript fully available?

Reviewer #2: Yes

Reviewer #3: Yes

Reviewer #5: Yes

5. Is the manuscript presented in an intelligible fashion and written in standard English?

Reviewer #2: Yes

Reviewer #3: Yes

Reviewer #5: Yes

6. Review Comments to the Author

Reviewer #2: The authors have incorporated the suggestions and I am completely satisfied with the answers. This manuscript is now acceptable.

Reviewer #3: The revised manuscript contains the necessary revisions. Therefore I am recommending it for publication.

Reviewer #5: In this paper, the authors are proposed “SARS-CoV-2 protein structure and sequence mutations: evolutionary analysis and effects on virus variantsSARS-CoV-2 protein structure and sequence mutations:”

The strengths of the paper are that it is well structured, the description of the related work is well done and that results are extensively compared to results of the similar research.

The all the reviewer comments has been addressed

7. PLOS authors have the option to publish the peer review history of their article (what does this mean?). If published, this will include your full peer review and any attached files.

Reviewer #2: **Yes: **Shaban Ahmad

Reviewer #3: **Yes: **Ishtiaque Ahammad

Reviewer #5: No

---

## [Editor Report · Acceptance letter]

11 Jul 2023

PONE-D-23-06693R1 

SARS-CoV-2 protein structure and sequence mutations: evolutionary analysis and effects on virus variants. 

Dear Dr. Guzzi:

I'm pleased to inform you that your manuscript has been deemed suitable for publication in PLOS ONE. Congratulations! Your manuscript is now with our production department. 

Kind regards, 

on behalf of

Dr. Nagarajan Raju 

Academic Editor

PLOS ONE